# Association of middle cerebral artery aneurysms and variation of the A1 segment

**Xiaohui Li[1,2], Xi Yue[2], Zhengyuan Xie[2], Lina Nie[2], Ge Huang[2], Yilong Peng[2], Jiyong Gu[2], Chan Lai[2], Hongzhi Gao[1¤]\***

1 Department of Neurosurgery, The Second Affiliated Hospital of Fujian Medical University, Quanzhou, Fujian, China, 2 Department of Neurosurgery, Jiangmen Central Hospital, Jiangmen, Guangdong, China

¤ Department of Neurosurgery, The Second Affiliated Hospital of Fujian Medical University, Quanzhou, Fujian, China.
\* gaohongzhi@fjmu.edu.cns

## Abstract

### Objective

The disturbance of blood flow caused by variations in the circle of Willis is an important factor in the occurrence and development of aneurysms. Previous studies have confirmed that a fetal-type posterior cerebral artery(PCA) is closely related to posterior communicating artery (PcoA) aneurysms, while anatomical variations of the anterior cerebral artery (ACA) appear to correlate with the prevalence of aneurysms in the anterior communicating artery (ACoA). However, the relationship between variations in the circle of Willis and middle cerebral artery(MCA) aneurysms remains controversial.

### Methods

This study retrospectively analyzed the Computed Tomography Angiography (CTA) data of 269 cases of patients with intracranial aneurysms and 269 cases of patients without aneurysms at the Jiangmen Central Hospital from January 2012 to December 2023. The 3D-Slicer software was utilized to measure the artery diameter and investigate the relationship between anatomical variations of the circle of Willis and MCA aneurysm.

### Results

In the aneurysm group, there were 39 cases of A1 dysplasia on the affected side, compared to 20 cases in the control group, with a significantly higher prevalence in the aneurysm group (P = 0.0125). The average diameter of middle cerebral arteries was smaller in the aneurysm group (2.304 ± 0.5613 mm) than in the control group (2.611 ± 0.5500 mm), showing a significant difference (P = 0.001).In aneurysm patients, the MCA diameter on the affected side was smaller in the A1 dysplasia group (2.156 ± 0.5256mm) compared to the A1 normal development group (2.405 ± 0.5718mm, P = 0.0114). Additionally, the average maximum aneurysm diameter was larger in the A1 dysplasia group (6.958 ± 5.163mm) than in the A1 normal development group (5.483 ± 3.336mm, P = 0.03).The presence of ipsilateral A1 dysplasia had a statistically significant effect on the occurrence and rupture of MCA aneurysms.

**Data availability statement:** All relevant data are within the manuscript and its Supporting Information files.

**Funding:** The author(s) received no specific funding for this work.

**Competing interests:** The authors have declared that no competing interests exist.

## Conclusions

The variation in the circle of Willis may impact the occurrence and rupture of MCA aneurysms by altering blood flow distribution, constricting the diameter of the parent artery, and shifting the location of blood flow impact.

## Introduction

Intracranial aneurysm is a common disease with a high mortality and disability rate in clinical practice. Its incidence rate accounts for approximately 1-3% of the population [1,2], rising to as high as 7% in the Asian population [1,3], and reaching up to 17% in high-risk groups [4]. Aneurysm rupture is the primary cause of poor prognosis, with a 30-day mortality rate of 45%, and 30% of survivors will experience moderate to severe disability [5]. Despite significant advancements in the diagnosis and treatment of aneurysms in recent years, the specific pathogenesis remains unclear [5]. Risk factors for aneurysms include smoking, alcohol abuse, hypertension, old age, family history, female gender, high doses of estrogen drug intake, cocaine abuse, among others [1,6]. Hemodynamic changes, such as the absence of components within the Circle of Willis, can lead to alterations in the direction and velocity of blood flow. These changes are also important factors in the occurrence and development of aneurysms. [7].

The circle of Willis is a group of arteries that connect the left and right sides of the brain, as well as the anterior and posterior circulation [8,9]. This structure exhibits significant variability, with the proportion of individuals possessing a complete circle of Willis varying widely in the literature, ranging from 12% to 60% [10–15]. Aneurysms are a common pathological occurrence in the circle of Willis [1], accounting for 90% of all aneurysms [5]. Kayembe et al [16]suggest that variations in the circle of Willis were more prevalent in individuals with aneurysms compared to those without. They proposed that hemodynamic changes resulting from variations in the circle of Willis were risk factors for the development and occurrence of aneurysms. These hemodynamic alterations, such as elevated wall shear stress(WSS) caused by changes in the circle of Willis, may serve as precipitating factors for aneurysm formation. The most common locations for aneurysms in the circle of Willis are the anterior communicating artery(AcoA), internal carotid artery(ICA), and middle cerebral artery(MCA) [1,17].Previous studies on the relationship between variations in the circle of Willis and the occurrence, development, and rupture of aneurysms have primarily focused on the fetal-type posterior cerebral artery(PCA) and posterior communicating artery (PcoA) aneurysms, as well as the anterior cerebral artery(ACA) A1 dysplasia and AcoA aneurysms [16,18–22]. However, there has been limited research on the variation of MCA aneurysms and their association with the circle of Willis. The study aims to investigate whether changes in the anatomy of the circle of Willis are associated with the occurrence and rupture of MCA aneurysms. In this study, we analyzed the CTA data of 269 patients with intracranial aneurysms and 269 patients without intracranial aneurysms at the Jiangmen Central Hospital from January 1, 2012, to December 31, 2023.

## Materials and methods

### 1. Study population and data collection

This study is a single-center retrospective study approved by the Ethics Committee of Jiangmen Central Hospital, with the batch number Jiangxin Medical Ethical Review (2024) no. 186A. As this was a retrospective study, the Ethics Committee agreed to waive the requirement for informed consent, and all methods were conducted in compliance with relevant guidelines and regulations. We had access to information that could identify individual participants during or after data collection for research purpose since July 25th, 2024. We systematically gathered data from hospitalized

patients diagnosed with saccular aneurysms over the period from January 1, 2012, to December 31, 2023, resulting in 1,146 cases of ruptured aneurysms and 662 cases of unruptured aneurysms..The inclusion criteria were as follows: [1] age over 18 years, [2] clear and artifact-free head CTA images from our hospital, and [3] CTA findings indicating a MCA aneurysm. Among the cases screened, there were 220 cases of ruptured aneurysms and 85 cases of unruptured aneurysms. Exclusion criteria were: [1] bilateral MCA aneurysms or multiple aneurysms in other locations, [2] intracranial space-occupying lesions such as brain tumors or parasites, [3] fusiform aneurysm,arteriovenous malformation, arteriovenous fistula, moyamoya disease, vasculitis, or dissection, [4] presence of ischemic stroke, intracranial vascular occlusion, or cerebral hemorrhage, or renal insufficiency, [5] acute or chronic infection, [6] previous intervention or surgery based on CTA examination, [7]Insufficient clear image data is caused by the lack of thin-slicer CTA data, which 3D-Slicer is unable to analyze, [8] unsuitability for enhanced CT examination. 269 cases of intracranial aneurysm patients were selected for the study. According to the age distribution of the aneurysm group, there were 5 cases in the 18-30 age range, 11 cases in the 31-40 age range, 58 cases in the 41-50 age range, 84 cases in the 51-60 age range, 77 cases in the 61-70 age range, 27 cases in the 71-80 age range, and 7 cases aged 81 or above. We selected 269 control cases, matched for age, from a pool of 7045 patients who underwent head CTA at the clinic during the same period and did not exhibit intracranial aneurysms, brain tumors, cerebral infarction, cerebral arteriovenous malformation, or other intracranial diseases.The selection process is outlined in Fig 1.

## 2. Image analysis

CTA was performed with a 64-row helical CT scanner (PHILIPS or TOSHIBA AQUILION 64) following the standard protocol at Jiangmen Central Hospital. All patients undergoing CTA (Dicom mode) were imported into 3D Slicer (version 5.4; Surgical Planning, Harvard University, Boston, MA, USA). The CTA-AAA2 Volume Rendering was used for rendering and reconstruction. The resulting 3D model can be rotated 360 degrees on the screen from any angle and the distance and size can be adjusted for research purposes. Regions of interest were delineated by visualizing the circle of Willis from various angles of each branch artery. Measurements of the diameter of the A1 segment of the ACA, M1 segment of the MCA, PCoA, P1 segment of the posterior cerebral artery, and AcoA were performed using the Markups module. Referring to the methods described in previous literature [19,23], the vessel diameter was measured in the middle third of the vessel perpendicular to the direction of the vessel. (Figs 2 and 3). A diameter of less than 0.8 mm was defined as dysplasia [24,25]. Cases of dysplasia of the A1, M1, AcoA, PcoA,P1 were counted. The diameter of aneurysms in the aneurysm group was also measured. Aneurysm size was determined based on the greatest diameter.The CTA sequences and the diameters of the 3D models were measured independently by a neurosurgeon (Yue Xi) and a chief radiologist (Lai Chan), with disputes resolved through group discussions.

## 3. Data statistics

The experimental data were entered using Excel 2013. All data were analyzed using SPSS 23.0 software (SPSS Inc., Chicago, Illinois). Measurement data that followed a normal distribution were presented as mean ± standard deviation (M±SD), and analyzed using t-tests and analysis of variance. For data that did not follow a normal distribution, the data were presented as median and quartile, and analyzed using non-parametric tests such as the Mann-Whitney test and Kruskal-Wallis test. Count data were presented as ratios, and statistical analysis was performed using the chi-square test or Fisher's exact test. In order to determine the independent predictors of A1 Dysplasia, multivariate logistic regression analysis was performed.A p-value of less than 0.05 was considered statistically significant.All tests were two-sided.

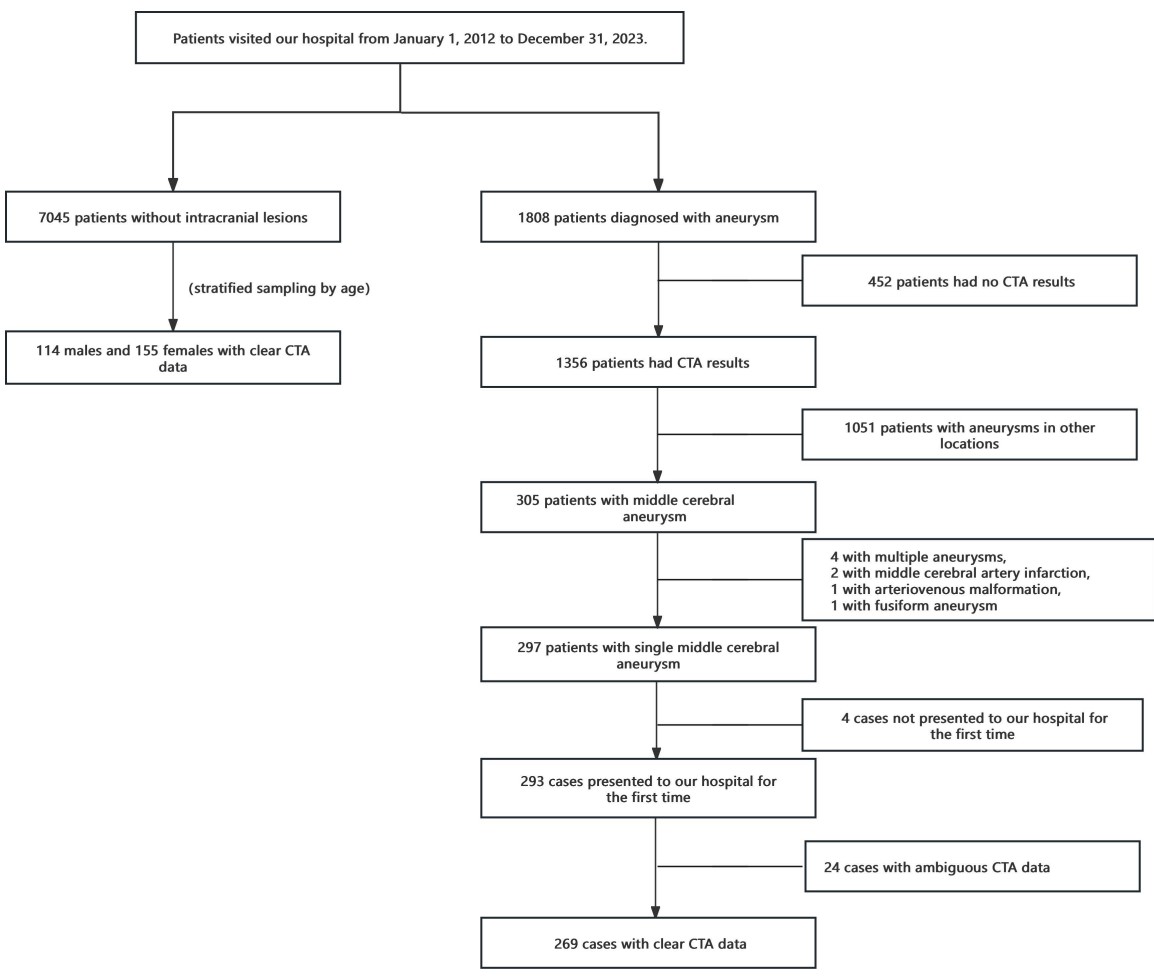

**Fig 1.** Enrollment flow chart.

## Results

### 1. Patient characteristics

A total of 269 patients with aneurysms and 269 age-matched controls were included in the study. The complete information regarding age, gender, and the circle of Willis is illustrated in Figs 4–6. In the aneurysm group,193 patients had a ruptured intracranial aneurysm, while 76 had an unruptured intracranial aneurysm. The aneurysms were further classified by location, with 115 cases on the left side and 154 on the right side.

### 2. Comparison of Dysplasia in the AcoA, PcoA, and PCA between the Aneurysm Group and Control Group

As shown in Table 1, there was no significant difference in the incidence of AcoA dysplasia between the two groups (P = 0.9221).There was no statistically significant difference in the incidence of PcoA dysplasia between the two groups (P=0.6031).There were differences in the incidence of PCA dysplasia among the three groups (P=0.0348). Pairwise comparison revealed no difference between unilateral PCA dysplasia and bilateral PCA dysplasia (P=0.3290), no difference between unilateral PCA dysplasia and normal PCA (P=0.1327), but a difference between bilateral PCA dysplasia and normal PCA (P=0.0352).

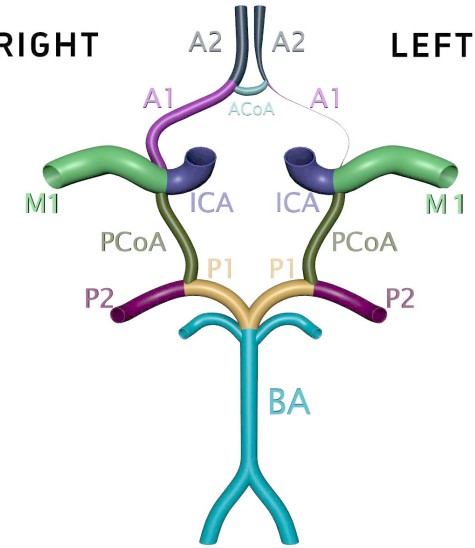

**Fig 2. Schematic diagram of each vascular component in the circle of Willis, showing A1 dysplasia on the left side.** A1 = A1 segment of the anterior cerebral artery, A2 = A2 segment of the anterior cerebral. artery, ACoA = anterior communicating artery, M1 = M1 segment of the middle cerebral artery, ICA = internal carotid artery, PCoA = posterior communicating artery, P1 = P1 segment of the posterior cerebral artery, P2 = P2 segment of the posterior cerebral artery, BA = basilar artery.

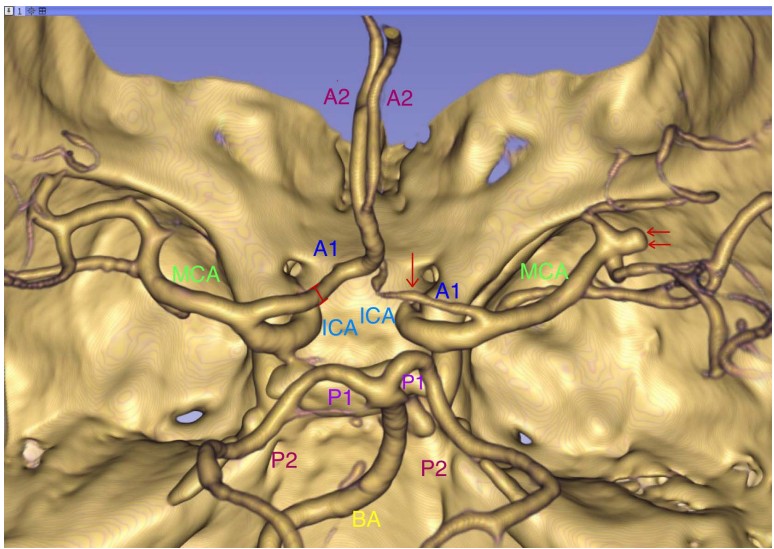

**Fig 3.** Example of image reconstruction. The single arrow indicates dysplasia of the A1 segment of the left anterior cerebral artery, while the double arrow indicates a MCA aneurysm. Abbreviations as in figure 2".

## 3. Comparison of A1 Dysplasia on the Affected Side between the Aneurysm Group and Control Group.

As shown in Fig 7, A1 dysplasia on the affected side was more common in the aneurysm group than in the control group (P=0.0125).

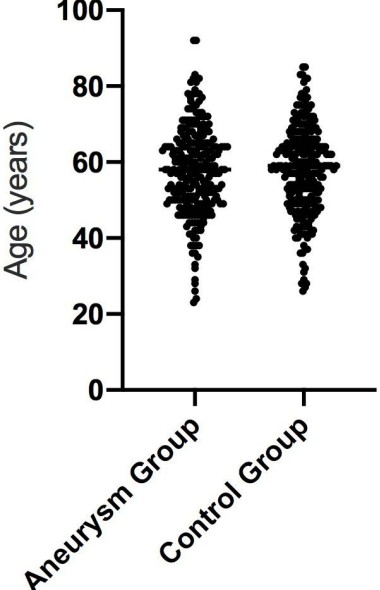

**Fig 4. The age distribution of both the aneurysm group and the control group.**

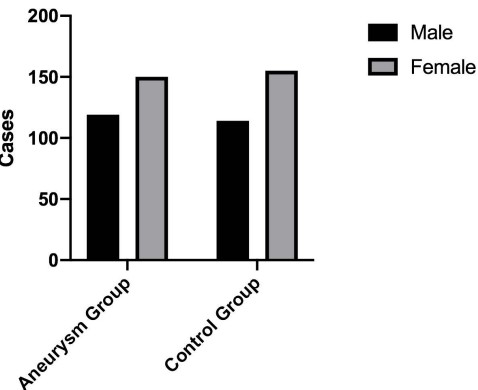

**Fig 5. The sex distribution in the aneurysm and control groups.**

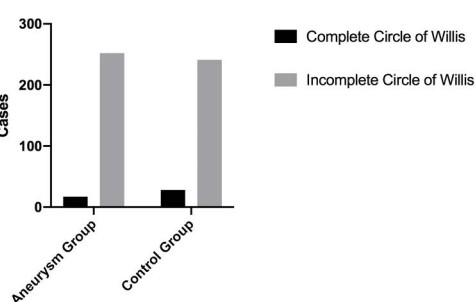

**Fig 6. The integrity of the circle of Willis in the aneurysm group compared to the control group.**

**Table 1. Data for aneurysm and control groups.**

| Variable | Aneurysm Group (n=269) | Control Group (n=269) | P-Value |
|---|---|---|---|
| Age (years) | 57.65±11.50 | 57.80±11.42 | 0.7520 |
| Gender (Male/female) | 119/150 | 114/155 | 0.7279 |
| Complete Circle of Willis | 17 | 28 | 0.1186 |
| AcoA dysplasia | 72 | 70 | 0.9221 |
| Unilateral PcoA dysplasia | 72 | 67 | |
| Bilateral PcoA dysplasia | 142 | 155 | 0.6031 |
| Bilateral PcoA Developed Normally | 55 | 47 | |
| Unilateral PCA dysplasia | 35 | 25 | |
| Bilateral PCA dysplasia | 17 | 7 | 0.0348 |
| Bilateral PCA Developed Normally | 217 | 237 | |
| A1 dysplasia | 39 | 20 | 0.0125 |
| Diameter of the M1 segment (mm) | 2.304±0.5613 | 2.611±0.5500 | 0.001 |

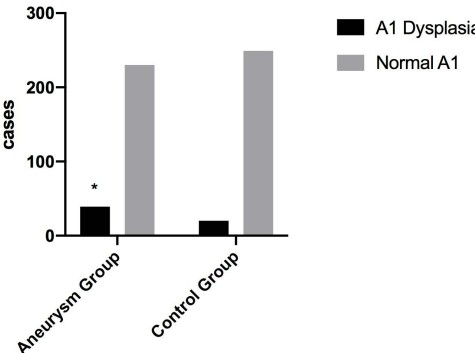

**Fig 7. The A1 dysplasia distribution in the aneurysm and control groups.**

## 4. Comparison of Diameter of the M1 segment between the Aneurysm Group and Control Group.

In the aneurysm group, the diameter of the M1 segment on the affected side was 2.280±0.5625, and on the contralateral side was 2.328±0.5601, with no significant difference between the two sides (P= 0.3252). In the control group, the diameter of the left M1 was 2.649±0.5521, and the right M1 was 2.572±0.5462, showing no difference between the two sides (P=0.1066). The overall diameter of all M1 arteries in the aneurysm group was 2.304±0.5613, while in the control group was 2.611±0.5500. The diameter of M1 arteries in the aneurysm group was smaller than in the control group (P = 0.001) (Fig 8).

## 5. Comparison of Diameter of the M1 segment between A1 Dysplasia and A1 Normal Groups in Patients with Aneurysms (Table 2)

In patients with aneurysms, the diameter of the M1 segment on the affected side in the A1 dysplasia group was 2.156 ± 0.5256mm, and on the contralateral side, it was 2.197 ± 0.4921mm. There was no significant difference between the two groups (P=0.7195). In the A1 normal development group, the M1 diameter on the affected side was 2.405 ± 0.5718mm, and on the contralateral side, it was 2.350 ± 0.5689mm. Again, there was no significant difference between the two groups (P=0.3012). The M1 diameter on the affected side in the A1 dysplasia group was smaller than in the A1 normal development group as shown in Table 2 (P=0.0114).

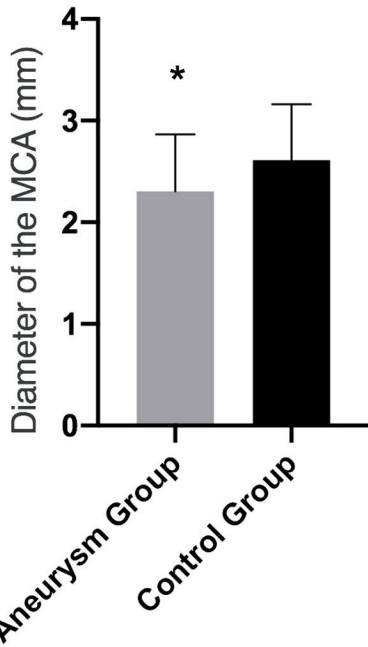

**Fig 8. The diameter of M1 arteries in the aneurysm and control groups.**

**Table 2. Comparison of M1 and max aneurysm diameters in A1 dysplasia vs. normal groups.**

| Variable | A1 dysplasia group | A1 Normal Development Group | P-Value |
|---|---|---|---|
| M1 diameter on the affected side | 2.156±0.5256 | 2.405±0.5718 | 0.0114 |
| Maximum Aneurysm Diameters | 6.958±5.163 | 5.483±3.336 | 0.0318 |

## 6. Comparison of Maximum Aneurysm Diameters between the A1 Dysplasia and A1 Normal Development Groups in Patients with Aneurysms (Table 2)

As shown in Table 2, the average maximum diameter of aneurysms in the A1 dysplasia group was larger (P<0.05).

## 7. A1 Dysplasia as a Predictor of Middle Cerebral Aneurysm Occurrence

Age, gender, smoking, drinking, hypertension, diabetes, and ipsilateral A1 dysplasia were included to construct a multivariate logistic regression equation. As shown in Table 3,the results indicated that hypertension had a statistically significant effect on the occurrence of MCA aneurysms (P<0.05). The impact of ipsilateral A1 dysplasia on the incidence of middle cerebral artery aneurysms was also statistically significant (P<0.05).

## 8. A1 Dysplasia as a Predictor of Middle Cerebral Aneurysm Rupture

Age, gender, smoking, drinking, hypertension, diabetes, maximum diameter of the aneurysm, and ipsilateral A1 dysplasia were included in constructing a multivariate logistic regression equation. As shown in Table 4, the results indicated that the impact of ipsilateral A1 dysplasia on middle cerebral artery aneurysm rupture was statistically significant (P<0.05).

**Table 3. Multivariate Logistic Regression of Variables for MCA Aneurysm Risk.**

| Variable | b Value | P Value | OR Value | 95% Confidence Interval |
|---|---|---|---|---|
| Age | -0.008 | 0.337 | 0.992 | 0.977-1.008 |
| gender* | -0.031 | 0.880 | 0.969 | 0.647-1.452 |
| smoking | 0.047 | 0.881 | 1.048 | 0.569-1.903 |
| drinking | -0.122 | 0.821 | 0.885 | 0.296-2.641 |
| hypertension | 0.627 | 0.009 | 1.871 | 1.179-2.970 |
| diabetes | -0.625 | 0.107 | 0.535 | 0.250-1.147 |
| ipsilateral A1 dysplasia | 0.812 | 0.006 | 2.252 | 1.257-4.037 |
| intercept | 0.164 | 0.724 | 1.178 | 0.475-2.922 |

*Women as controls

**Table 4. Multivariate Logistic Regression on MCA Aneurysm Rupture.**

| Variable | b Value | Wald Value | P Value | OR Value | 95% Confidence Interval |
|---|---|---|---|---|---|
| Age | -0.005 | 0.157 | 0.692 | 0.995 | 0.970-1.021 |
| gender* | -0.300 | 0.926 | 0.336 | 0.741 | 0.403-1.364 |
| smoking | -0.190 | 0.164 | 0.685 | 0.827 | 0.330-2.072 |
| drinking | 0.732 | 0.864 | 0.353 | 2.078 | 0.444-9.720 |
| hypertension | -0.329 | 1.211 | 0.271 | 0.720 | 0.401-1.293 |
| diabetes | -1.068 | 3.795 | 0.051 | 0.344 | 0.117-1.007 |
| Maximum Aneurysm Diameters | 0.036 | 0.620 | 0.431 | 1.037 | 0.948-1.134 |
| ipsilateral A1 dysplasia | 1.242 | 5.009 | 0.025 | 3.461 | 1.167-10.267 |
| intercept | 1.299 | 2.608 | 0.106 | 3.667 | |

*Women as controls.

## Discussion

In our study, we selected 1356 patients with CTA data from a total of 1808 hospitalized patients with aneurysms. Ensuring the uniform use of CTA data and excluding cases with only MRA data helped maintain the homogeneity of the study. While most of the research have predominantly concentrated on the correlation between ACA A1 segment variations and AcoA aneurysms, as well as the association between fetal-type PCA and PcoA aneurysms [16,18–23,26–28].There have been few studies on the connection between MCA aneurysms and variations in the circle of Willis, with some suggesting that these variations do not impact the occurrence of MCA aneurysms [23,29]. Therefore, among the 1356 cases of CTA data, we specifically selected patients with MCA aneurysms only for this study. Patients with other diseases that could potentially affect arterial flow were excluded from the analysis. Additionally, we omitted patients who had initially presented to other hospitals to avoid any potential confounding factors related to changes in the blood vessel shape resulting from prior treatments for the aneurysm.However, our study found that the incidence of A1 dysplasia was higher in the aneurysm group compared to the control group, indicating that variations in the circle of Willis may also contribute to the development of MCA aneurysms.

The variation in the Circle of Willis is a common phenomenon within the population, and its integrity rate often varies across different studies due to differing classification methods [30]. CTA is more effective than Digital Subtraction Angiography (DSA) in visualizing normally sized arteries [31]. Studies utilizing CTA have reported Circle of Willis completeness rates

ranging from 15.3% to 37.2% [13,32,33]. In our dataset, the completeness rate of the Circle of Willis in individuals with aneurysms was 6.3%, compared to 10.4% in the control group. The overall completeness rate of 8.4% observed in our study is lower than previously reported rates.The rationale may encompass two aspects.First, age is an important factor to consider, as the average age of aneurysm onset is typically higher than that of the general population. The control group was selected to match the age distribution of the aneurysm group. Therefore, both the aneurysm group and the control group had higher average ages compared to the normal population.However, individuals under the age of 40 have a more complete Willis circle [34]. Second, we employed a criterion that considered an artery diameter of less than 0.8 mm as absent, which may have led to an underestimation of the actual completeness rate. A straightforward comparison between the aneurysm group and the control group revealed no significant differences in the variations of the Circle of Willis in our study.The study did not further differentiate the component blood vessels of the Circle of Willis, potentially due to the assumption that mutations in the Circle of Willis identified in a previous study are not significantly related to the development of the MCA aneurysms.

Saccular intracranial aneurysms are believed to arise from a cystic bulge in the arterial wall, which is induced by hemodynamic stress and a subsequent inflammatory response [15]. Kayembe K. N. et al.hypothesized that the elevation in WSS induced by variations in the Circle of Willis may contribute to aneurysm formation [16]. Furthermore, they identified hemodynamic alterations resulting from variations in the Circle of Willis as significant risk factors for the onset and progression of aneurysms. [16,35] .Numerous studies have established a significant correlation between elevated WSS and high oscillatory shear index (OSI) with the occurrence and progression of aneurysms, as evidenced by animal experiments [36], dynamic research models [17], and human studies [37]. We hypothesize that variations in the Circle of Willis may influence the incidence of MCA aneurysms. Our study also found that the maximum diameter of aneurysms in the A1 dysplasia group was larger than that in the A1 normal development group. The potential mechanisms underlying this hypothesis can be summarized in three points.

Firstly, variations in the Circle of Willis may lead to alterations in cerebral blood flow distribution, deviating from the patterns observed in a normal Circle of Willis [38].Both excessive and insufficient blood flow stimulation of the vessel wall can lead to pathological alterations in the biomechanical properties of the arterial wall. Additionally, pathological high WSS resulting from variations in the Circle of Willis can induce morphological and functional changes in the endothelium within regions of blood flow disturbance in vivo [39,40]. Complex blood flow patterns are hypothesized to enhance inflammatory cell infiltration in the aneurysm wall [41]. Blood flow conditions can cause high WSS to activate pro-inflammatory signals in endothelial cells, which in turn recruit macrophages to sites exposed to high WSS, specifically through macrophage chemoattractant protein 1 (MCP1). The infiltration of macrophages leads to the expression of proteases, disruption of the internal elastic layer and collagen matrix, ultimately resulting in the formation of focal exocele and initiation of intracranial aneurysm [42].These modifications in endothelial phenotypes are believed to contribute to the localized progression of atherosclerotic risk factors [43,44].When one side of the A1 segment is dysplasia, there is an increase in the blood flow to both A2 segments via the contralateral ACA supply [45]. Consequently, the blood flow in the A1 segment and the AcoA undergoes dynamic remodeling, resulting in a significant increase in the blood flow within the AcoA [26]. This mechanism contributes to the formation of AcoA aneurysms [27].Hendrikse J et al. utilized phase-contrast magnetic resonance angiography to determine a volume flow of 245 ± 65mL/min in the ICA of subjects in the normal circle of Willis. However, in the group with A1 segment deletion, the contralateral ICA blood flow (303±56 mL/min) was significantly

increased, while the ipsilateral blood flow (214 ±94mL/min) was correspondingly decreased [46]. Zheng R et al. demonstrated in their experiment that in the absence of A1, the blood flow in the AcoA significantly increased, leading to a noticeable enlargement of the AcoA, confirming dynamic vessel reshaping [26]. All of these findings explain the increased blood flow in the AcoA when A1 is deficient. However, they overlook the fact that the blood flow in the ipsilateral ICA primarily supplies the ipsilateral MCA, potentially resulting in a higher flow in the MCA on the A1 deficient side compared to the contralateral MCA. This was directly illustrated in the hemodynamic studies conducted by Fahy P et al. using a 3D visualization model of the circle of Willis [47]. In normal configurations of the circle of Willis, the blood flow peak for one side of MCA arteries equated to 30%. Conversely, in the absence of the A1 segment, the same side's MCA could receive up to 34.6% of the flow, compared to 28.6% on the non-lacking side. Based on this, it can be inferred that ipsilateral A1 dysplasia may lead to an increased blood flow in the ipsilateral MCA relative to the contralateral and normal MCA of the circle of Willis. This could subsequently elevate WSS and contribute to a heightened incidence of MCA aneurysms. Lazzaro MA et al. primarily investigated the correlation between A1 deletion and AcoA aneurysms, as well as the association between fetal-type PCA and PcoA aneurysms [23]. They believe that excessive blood flow from the ICA may not result in abnormal hemodynamic pressure on the MCA bifurcation in patients with a fetal-type PCA. However, they did not investigate excessive blood flow in the ipsilateral MCA with A1 dysplasia.

Secondly, in addition to the increase in blood flow in the ipsilateral MCA, unilateral A1 dysplasia may also reduce the diameter of the ipsilateral MCA due to atherosclerosis.Previous studies have shown that incomplete Wills circle affect the distribution of atherosclerotic plaques in the MCA [48]. This alteration impacts the WSS and oscillatory shear index (OSI) of the MCA, thereby influencing the development and occurrence of aneurysms. Previous research has suggested a relationship between systemic atherosclerosis and cerebral artery diameter [49]. OSI measures the degree and variability of oscillatory flow in the direction of shear stress, with high OSI values being linked to aneurysm formation and progression. Increasing the diameter of blood vessels directly decreases the aneurysm sac on the surface in relation to OSI values. Conversely, the WSS of the average aneurysmal vessel wall decreases as the parent vessel diameter increases [50]. Our study did not find any significant difference in the diameter of the left and right MCA in the aneurysm group. The diameter of the MCA in the MCA aneurysm group was smaller than in the control group, consistent with previous findings [51]. However, upon comparing the diameter of the MCA between the A1 dysplasia group and the A1 normal group, we observed that the ipsilateral MCA in the A1 dysplasia group was smaller than the ipsilateral MCA in the A1 normal group.

Furthermore, within the intact Circle of Willis, the location of the flow impact coincides with the site of multiple aneurysms [47]. The initial point of blood flow impact from the ICA is at the intersection of the ICA and the PCoA, where the incidence of PCoA aneurysms is also the highest [52]. However, in cases where the A1 segment is dysplasia, the impact point of blood flow from the ICA may extend to the bifurcation of the MCA. The posterior shift of the impact bifurcation point, potentially caused by variations in the Circle of Willis [15], may also contribute to the increased incidence of MCA aneurysms resulting from A1 dysplasia.

Given that the flow of the posterior circulation typically influences MCA flow via the PcoA artery, coupled with the observation that the incidence of PCoA dysplasia did not significantly differ between the aneurysm and control groups, it is plausible to infer that PCA dysplasia is more likely a manifestation of systemic atherosclerosis rather than a factor influencing the development of MCA aneurysms.

Limitations: Our study revealed a higher prevalence of A1 dysplasia in patients with MCA aneurysms compared to the control group, indicating a possible connection between specific variations in the circle of Willis and the pathogenesis of MCA aneurysms. Furthermore, we found that A1 dysplasia can serve as a predictor for the occurrence and rupture of middle cerebral aneurysms. These findings can guide decisions regarding the prevention and early treatment of unruptured aneurysms.However, we have only demonstrated a co-existence, not a direct causal relationship.The use of hydrodynamic simulation to investigate the impact of A1 dysplasia on the pathogenesis of MCA aneurysms is a commonly employed research method and should be the future research focus of our group.Furthermore, it would strengthen our findings to track the incidence of MCA aneurysms in individuals with A1 dysplasia over time. However, given that A1 dysplasia is present in only 0.87% of the population [53] and its definition varies, with criteria including less than 50% contralateral diameter [45], less than 0.8 mm [24,25,54], less than 1 mm [13,33], and classification based on contralateral A1 [19], further long-term research is needed in this prospective study.Lastly, in the population selection of aneurysms, a higher proportion of patients in our case group had ruptured aneurysms, as individuals with ruptured aneurysms have a greater urgency to seek medical treatment. Conversely, the general population typically exhibits a higher proportion of unruptured aneurysms. Additionally, blood vessel spasms caused by a ruptured aneurysm may affect the size of the artery. This discrepancy could potentially impact our results.Furthermore, this was a single-center retrospective study,all cases originated from patients admitted to our hospital, which could lead to bias in patient selection. The results of this study will need to be validated in a larger multicenter study.

## Conclusion

Hemodynamic changes resulting in variations within the Circle of Willis are identified as risk factors for the onset and rupture of aneurysms. Such variations potentially influence not only the formation of AcoA aneurysms and ICA aneurysms, but may also contribute to the development of MCA aneurysms. This exploratory study generates the hypothesis that longstanding A1 changes could correlate to MCA aneurysms, which should be the subject of a subsequent, more robust investigation.The underlying mechanism is believed to involve modifications in the distribution of arterial blood flow, parent artery atherosclerosis leading to diameter constriction, and alterations in the impact points of blood flow.

**Acknowledgments:** Thank Jinhong Li guidelines for the submission process, JinlangHe, Xiaolong Yuan, Xianming Deng support for data collection.

## Author contributions

**Conceptualization:** Xiaohui Li, hongzhi Gao.

**Data curation:** Xi Yue, Lina Nie, Chan Lai.

**Formal analysis:** Zhengyuan Xie.

**Funding acquisition:** Ge Huang, hongzhi Gao.

**Investigation:** Xiaohui Li, Lina Nie.

**Methodology:** Xiaohui Li.

**Project administration:** Xiaohui Li.

**Resources:** Ge Huang, Jiyong Gu.

**Software:** Zhengyuan Xie.

**Supervision:** Jiyong Gu, hongzhi Gao.

**Validation:** Zhengyuan Xie.

**Visualization:** Yilong Peng.

**Writing – original draft:** Xiaohui Li.

**Writing – review & editing:** hongzhi Gao.

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
