## [Decision Letter · Decision Letter 0]

24 Sep 2024

PONE-D-24-36671Association of middle cerebral artery aneurysms and variation of the A1 segmentPLOS ONE

Dear Dr. Gao,,

Thank you for submitting your manuscript to PLOS ONE. After careful consideration, we feel that it has merit but does not fully meet PLOS ONE’s publication criteria as it currently stands. Therefore, we invite you to submit a revised version of the manuscript that addresses the points raised during the review process.

We look forward to receiving your revised manuscript.

Kind regards,

Atakan Orscelik

Academic Editor

PLOS ONE

Reviewers' comments:

Reviewer's Responses to Questions

**Comments to the Author**

1. Is the manuscript technically sound, and do the data support the conclusions?

Reviewer #1: Yes

Reviewer #2: Yes

Reviewer #3: Yes

Reviewer #4: Yes

Reviewer #5: Partly

Reviewer #6: Partly

Reviewer #7: No

Reviewer #8: No

Reviewer #9: Yes

2. Has the statistical analysis been performed appropriately and rigorously? 

Reviewer #1: Yes

Reviewer #2: Yes

Reviewer #3: Yes

Reviewer #4: Yes

Reviewer #5: Yes

Reviewer #6: No

Reviewer #7: No

Reviewer #8: N/A

Reviewer #9: Yes

3. Have the authors made all data underlying the findings in their manuscript fully available?

Reviewer #1: Yes

Reviewer #2: No

Reviewer #3: Yes

Reviewer #4: Yes

Reviewer #5: Yes

Reviewer #6: Yes

Reviewer #7: Yes

Reviewer #8: Yes

Reviewer #9: Yes

4. Is the manuscript presented in an intelligible fashion and written in standard English?

Reviewer #1: Yes

Reviewer #2: Yes

Reviewer #3: No

Reviewer #4: Yes

Reviewer #5: Yes

Reviewer #6: Yes

Reviewer #7: Yes

Reviewer #8: No

Reviewer #9: Yes

5. Review Comments to the Author

Reviewer #1: There are certain editions, which I would suggest prior to accepting the manuscript in its current form.

1. The authors must describe their methodology in detail- in regards to reason for exclusion of each section as per their flow diagram. This reasoning can be added to the discussion section.

2. Some demographic results are duplicated in the description. These need to be removed. Page 7 line 129-139. The authors can focus on describing positive highlights from these results table in description.

3. Table 4 and 5 must include the numerical values and percentages of parameters compared between the two groups. Similarly in other statistical tables.

Decision: Minor revision

Reviewer #2: As a proficient writer, I appreciate your thorough evaluation of the relationship between vascular variation and aneurysm rupture, as well as its subsequent exacerbation. However, I noticed that you did not provide a comprehensive description of how vascular variation is measured; this aspect could be elaborated upon. Furthermore, regarding the final discussion on mechanisms, there has been insufficient exploration of the abnormalities in various blood vessels and their role in aneurysm pathogenesis. Potential mechanisms such as inflammation, immune responses, mutations, or alterations in cerebral vascular circulation warrant further consideration for a more nuanced explanation.

Reviewer #3: The study investigates the relationship between variations in the circle of Willis and middle cerebral artery (MCA) aneurysms.

It shows higher prevalence in the aneurysm group of A1 dysplasia in the affected side, smaller average diameter of middle cerebral arteries, MCA diameter on the affected side smaller in the A1 dysplasia group (2.156 ± 0.5256mm) compared to the A1 normal development group (2.405 ± 0.5718mm, P = 0.0114). Furthermore, the average maximum aneurysm diameter was larger in the A1 dysplasia group. The presence of ipsilateral A1 hypoplasia had a statistically

significant effect on the occurrence and development of MCA aneurysms.

The study therefore suggests that the variation in the circle of Willis may impact the occurrence and progression of MCA aneurysms by altering blood flow distribution, constricting the diameter of the parent artery, and shifting the location of blood flow impact.

The study cohort is large, the methods are sound, but the results description is sloppy and difficult to follow for unnecessary repetition of data in the text and in the tables. The figures are useless repetitions of data already in the text and in the tables.

Other points to be amended:

- Line 59: Padget is not referenced.

- A schematic figure reporting anatomy and names of the circle of Willis may be useful for the non-neurologists.

- Line 78-79: the sentence is unclear.

- Table 1 is better indicated as a figure.

- The imagine scheme is a recipe and not a method description.

- Line 113: hypoplasia should be better defined and its difference from dysplasia clarified.

- Figure 1: name of arterial segments and indication of hypoplasia should be added with arrows.

- Table 2: the number of subjects should be added in brackets for an easy calculation of fractions.

- Results sections: discuss only the significant differences, cancel useless figures and concentrate the numbers in appropriate tables.

- Tables need a heading.

- The discussion starts with an unnecessary repetition of the introduction.

- Line 232: posited? Or hypothesized?

- Line 288: I suggest “limitations”.

- Reference 9 is not in a standard format.

Reviewer #4: This is a retrospective study designed to look at the association of A1 segment dysplasia on occurrence of MCA aneurysms.

Research methodology and writing are good.

However the manuscript does require revisions for consideration for publication. Revisions for consideration should include-

Line 41 - We cannot show that there is an effect on development or causation in a retrospective cross sectional study, I think the authors intended to use the word rupture but used development.

Line 55 - What do the authors mean by " blood flow disorders". Needs elucidation.

Line 70 - We already know variations in Circle of Willis are associated with increased incidence of intracranial aneurysms. The statement should have said " incidence of middle cerebral artery aneurysms" not intracranial aneurysms in general.

Line 89 - Please elucidate the methodology for selecting the controls.

Line 202 - Dysplasia as a predictor of " Rupture" not "development".

Line 224 and 225- The reasoning for lower rate of complete Circle of Willis in the population being studied has been described but does not make sense. If age of onset of aneurysms is higher and the completeness rate of Circle of Willis is lower in younger individuals, shouldn't the completeness rate of circle of Willis be higher in this study group rather than being lower, as this is a study directed at studying patients with aneurysms who are in general older.

Line 263 - The statement regarding embryonic ICA does not make sense. This needs clarification.

Along with the corrections mentioned above, there seems to be predominance of ruptured aneurysm patients in this study compared to general population which usually has more patients with unruptured aneurysms. This limitation should be acknowledged.

Reviewer #5: 1.It appears that the A1 segment of the ACA might be associated with a smaller MCA diameter which might be associated with the formation of MCA aneurysms. However, at the end of the study, the writer listed several possible confounders to include possible selection bias.

2.I am curious as to what constituted for "lack of clear image data, severe artifacts" as can be seen in line 87. This could be subjective to the radiologists or neurologists looking at the study. Or the researchers who conducted the study. This goes back to the possibility of selection bias.

3.clearer distinction between causality and co-existence needed.

4.further distinction between the types of aneurysms may be helpful in substantiating the proposed mechanisms of such aneurysmal formation.

Reviewer #6: First, in the article the authors counted patient data spanning the corresponding CTA examinations done between 2012 and 2023, but Jiangmen Central Hospital only introduced the 64-row spiral CT described in the article in 2020, so did all the data on patients with aneurysms included in the study within the present study originate after the introduction of the 64-row CT or before? The authors should provide an explanation;

Secondly, the interpretation of the CTA results was performed by a young neurosurgeon and a senior imaging physician; this arrangement reduces the accuracy of the CTA results even though both physicians independently interpreted the CTA results; two senior imaging or neurosurgeons should have performed the independent interpretation, which would have been more convincing, and the results would have been more accurate and credible. The results should be more convincing and the results more accurate and credible.

Once again, the authors were slightly confused when analyzing the corresponding results for comparisons between different types of subgroups; it was not clear what the p-value was when applying the statistical analysis for a statistically significant comparison between the two;

Meanwhile, for the corresponding results of themultivariate logistic regression analysis, the results were not displayed with common graphs;

In the discussion section, the authors did not discuss and analyze the results.

Reviewer #7: Dear authors

I would to ask you:

- how did you pair your samples, by convenience only? can you show more data (besides age and sex) so we can understand how good is the matching? have you estimated any sample size?

- please add the rupture status and vasospasm occurrence in the analysis, since they are a major source of vessel diameter variation and are expected to be very different among the cases and controls?

- the vessel measurements were made on VRT images only (as suggested by Figure 1)? how did you accounted for the variability of size regarding the fitting of the opacification curves?

- have you re-assessed the vessel diameter after the vasospasm phase in ruptured aneurysms?

Sincerely

Reviewer #8: 1.The purpose of this study is to explore the relationship between middle cerebral artery aneurysms and anterior cerebral arteries. What are the types of middle cerebral artery aneurysms included here? (M1 segment? bifurcation? Or M2 segment)

2. The patency of the willis loop has a significant impact on cerebral blood flow. Why is this study only focused on the relationship between the anterior and middle cerebral aneurysms in the willis? According to the author's viewpoint, the absence of other blood vessels in the Willis loop may also affect the formation of cerebral aneurysms.

3.In the article's methodology, it is mentioned that there is also a control group, and how the control group is selected should be detailed and reflected in the overall flowchart.

4.In the result part（line 204）,”The results indicated that the impact of ipsilateral A1 dysplasia on middle cerebral artery aneurysm rupture was statistically significant”. However， the purpose of the article is to confirm the impact of ipsilateral A1 dysplasia on middle cerebral artery aneurysm. The data results did not directly confirm the author's viewpoint.

Reviewer #9: Authors of the current study reported association of A1 hypoplasia with MCA aneurysm and rupture. Theoretically abnormality of intracranial vessels would affect hemodynamic flow and subsequently lead to aneurysm formation and growth. Authors proved this association with A1 hypoplasia and MCA aneurysm. I have no comments as the analysis was appropriate with appropriate statistical tools. Would recommend authors to include low power as a limitation of the study and further the results have to be validated on a larger multicenter study. Further the clinical significance of finding is missing in the paper. Would recommend authros to add this.

6. PLOS authors have the option to publish the peer review history of their article (what does this mean? ). If published, this will include your full peer review and any attached files.

**Do you want your identity to be public for this peer review?** For information about this choice, including consent withdrawal, please see our Privacy Policy .

Reviewer #1: **Yes: ** SanjeevSreenivasan

Reviewer #2: **Yes: ** Li Yunze

Reviewer #3: **Yes: ** Andrea Semplicini

Reviewer #4: No

Reviewer #5: **Yes: ** Suhas Gangadhara

Reviewer #6: No

Reviewer #7: No

Reviewer #8: No

Reviewer #9: No

---

## [Author Response · Author response to Decision Letter 1]

7 Oct 2024

On behalf of my co-authors, we thank you for giving us the opportunity to revise and improve the quality of our article. We have carefully reviewed the comments from both the reviewers and yourselves, and have made revisions which are marked in red within the paper. We have made every effort to address the comments provided. Please find attached the revised version, which we would like to submit for your kind consideration.

---

## [Decision Letter · Decision Letter 1]

28 Oct 2024

PONE-D-24-36671R1Association of middle cerebral artery aneurysms and variation of the A1 segmentPLOS ONE

Dear Dr. Gao,

Thank you for submitting your manuscript to PLOS ONE. After careful consideration, we feel that it has merit but does not fully meet PLOS ONE’s publication criteria as it currently stands. Therefore, we invite you to submit a revised version of the manuscript that addresses the points raised during the review process.

We look forward to receiving your revised manuscript.

Kind regards,

Atakan Orscelik

Academic Editor

PLOS ONE

**Journal Requirements:**

Reviewers' comments:

Reviewer's Responses to Questions

**Comments to the Author**

1. If the authors have adequately addressed your comments raised in a previous round of review and you feel that this manuscript is now acceptable for publication, you may indicate that here to bypass the “Comments to the Author” section, enter your conflict of interest statement in the “Confidential to Editor” section, and submit your "Accept" recommendation.

Reviewer #1: All comments have been addressed

Reviewer #2: All comments have been addressed

Reviewer #3: All comments have been addressed

Reviewer #6: All comments have been addressed

Reviewer #7: (No Response)

Reviewer #9: All comments have been addressed

2. Is the manuscript technically sound, and do the data support the conclusions?

Reviewer #1: Yes

Reviewer #2: Yes

Reviewer #3: Yes

Reviewer #6: Yes

Reviewer #7: Partly

Reviewer #9: Yes

3. Has the statistical analysis been performed appropriately and rigorously? 

Reviewer #1: Yes

Reviewer #2: Yes

Reviewer #3: Yes

Reviewer #6: Yes

Reviewer #7: No

Reviewer #9: Yes

4. Have the authors made all data underlying the findings in their manuscript fully available?

Reviewer #1: Yes

Reviewer #2: Yes

Reviewer #3: Yes

Reviewer #6: Yes

Reviewer #7: Yes

Reviewer #9: Yes

5. Is the manuscript presented in an intelligible fashion and written in standard English?

Reviewer #1: Yes

Reviewer #2: Yes

Reviewer #3: No

Reviewer #6: Yes

Reviewer #7: Yes

Reviewer #9: Yes

6. Review Comments to the Author

**Reviewer #1:**  The authors have made the necessary changes and the manuscript can be accepted in its current format for publication.

Thank you

**Reviewer #2:**  The objective of this paper is commendable，it utilizes clinical data for publication and possesses the necessary ethical approval documents, making it suitable for dissemination.

**Reviewer #3: ** The authors have accepted all the comments and have implemented the necessary corrections. However, there are still inconsistencies and errors that need a careful revision by an English-speaking scientific expert.

They can be enlisted as follows:

- Line 115, a verb is missing.

- Figures 4 and 6 captions are awkward.

- Heading of table 1 is too descriptive.

- Description of column 1 in table 1 need rewording.

- Heading of table 2, 3, and 4 is too descriptive. These comments should stay in the text.

- There is no consistency between abbreviations in fig. 2, 3 and text.

- Letters of P2 and MCA in figure 2 are too clear to be recognized.

- The sentence on line 220 is unclear.

**Reviewer #6: ** The author responded to each of the corresponding questions I asked, but two of them were not convincingly answered. Therefore, I believe that the article needs to be further revised to meet the criteria for publication in your journal.

**Reviewer #7: ** Dear Authors

I read the revised version of the manuscript. You addressed some issues; core questions remain, however:

1) This is a cross-sectional study. Therefore, it is not scientifically possible to establish a temporal correlation between "dysplasia" and "aneurysm development". So, make your work consistent with the methodology and change your writing to "correlated", "associated", and the like.

2) The occurrence of subarachnoid hemorrhage is at the core of your investigation! Seventy-two percent of the aneurysm group had it. Nearly half of these patients will have temporarily false measures of the vessels' diameter - A1 included. Since you have not made a follow-up imaging, you have two ways to solve this: a. include the ruptured status in the multivariate analysis and discuss accordingly. Prevalence-wise, your findings will likely correlate to ruptured status. Then, discuss how this impacts the results. Or b. discuss that your methodology is insufficient to provide clear conclusions about baseline vessel assessments - therefore, you cannot exclude that the results can be biased. Accordingly, your conclusion should be that this exploratory study generates the hypothesis that longstanding A1 changes could correlate to MCA aneurysms, which should be the subject of a subsequent, more robust investigation.

3) The prevalence of A1 dysplasia in the aneurysm group was 39/269 (0.1449). The prevalence of A1 dysplasia in the control group was 29/269 (0.1078). The reported p-value (P = 0.0125) is not reproducible - neither by the two-sided Fisher nor the Chi-square tests. I found the significance levels varied from 0.2428 to 0.1945. For the suggested level of significance, you would need at least 1,263 subjects. If this is not the case, please explain and attach you statistical calculations to clarify.

4) You replied about the measurement methodology question, and cited two articles. The cited studies does not backup your claims. If you take a closer look at the Krasny (DOI:10.1136/neurintsurg-2013-010669) and Lazzaro (DOI:10.1136/jnis.2010.004358), both used planar DSA as the reference method. Interestingly, those authors address the limitations of CTA in assessing the vessels diameters. My suggestions are: a. find the appropriate references that back up your claims, but be mindful that 95% of reader are convinced that DSA is still the gold standard, as medication societies support. Or b. state that you developed this methodology and discuss the limitations.

Sincerely

**Reviewer #9: ** (No Response)

7. PLOS authors have the option to publish the peer review history of their article (what does this mean? ). If published, this will include your full peer review and any attached files.

**Do you want your identity to be public for this peer review?** For information about this choice, including consent withdrawal, please see our Privacy Policy .

Reviewer #1: No

Reviewer #2: No

Reviewer #3: **Yes: ** Andrea Semplicini MD

Reviewer #6: No

Reviewer #7: No

Reviewer #9: No

---

## [Author Response · Author response to Decision Letter 2]

8 Nov 2024

Dear Editors,

On behalf of my co-authors, we thank you for giving us the opportunity to revise and improve the quality of our article. We have carefully reviewed the comments from both the reviewers and yourselves, and have made revisions which are marked in red within the paper. We have made every effort to address the comments provided. Please find attached the revised version, which we would like to submit for your kind consideration.

In accordance with your request, I have ensured that the citations are accurate and complete, and that no references have been made to retracted papers. Thank you.

We have prepared a detailed response letter addressing each of the reviewer comments, which can be found at the end of this letter. We found the comments to be extremely helpful and would like to express our gratitude to both you and the reviewers for your valuable feedback.

We hope that this revised version meets the standards for publication in your journal. Thank you for considering our submission, and we look forward to hearing from you soon.

Sincerely,

Hongzhi Gao

Reviewer #1: The authors have made the necessary changes and the manuscript can be accepted in its current format for publication.

Author response:Thank you！

Reviewer #2: The objective of this paper is commendable，it utilizes clinical data for publication and possesses the necessary ethical approval documents, making it suitable for dissemination.

Author response:Thank you！

Reviewer #3: The authors have accepted all the comments and have implemented the necessary corrections. However, there are still inconsistencies and errors that need a careful revision by an English-speaking scientific expert.

They can be enlisted as follows:

- Line 115, a verb is missing.

Author response:Thank you for the correction. We have added the verb.

- Figures 4 and 6 captions are awkward.

- Heading of table 1 is too descriptive.

- Description of column 1 in table 1 need rewording.

- Heading of table 2, 3, and 4 is too descriptive. These comments should stay in the text.

Author response:Thank you for your corrections, we have revised the manuscript according to your suggestions.

- There is no consistency between abbreviations in fig. 2, 3 and text.

- Letters of P2 and MCA in figure 2 are too clear to be recognized.

Author response:Thank you for your comment. I have redrawn a clearer Figure 2 and added annotations for the abbreviations.

- The sentence on line 220 is unclear.

Author response:Thank you for the correction. I have reorganized the sentence in line 220 to make it more coherent.

Reviewer #6: The author responded to each of the corresponding questions I asked, but two of them were not convincingly answered. Therefore, I believe that the article needs to be further revised to meet the criteria for publication in your journal.

Author response:Thank you very much for your constructive criticism. However, I am sorry to say that you did not specify which two issues did not convince you, so I am unable to make further revisions. Thank you!

Reviewer #7: Dear Authors

I read the revised version of the manuscript. You addressed some issues; core questions remain, however:

1) This is a cross-sectional study. Therefore, it is not scientifically possible to establish a temporal correlation between "dysplasia" and "aneurysm development". So, make your work consistent with the methodology and change your writing to "correlated", "associated", and the like.

Author response:Thank you for your comment. This retrospective study begins with two groups, one with the middle cerebral artery aneurysms (cases) and one without (controls). Then we look back in time to assess exposure history in both groups.So we believe our study is a case-control study, which allows us to assess a possible causal relationship between the disease and exposure factors.

2)The occurrence of subarachnoid hemorrhage is at the core of your investigation! Seventy-two percent of the aneurysm group had it. Nearly half of these patients will have temporarily false measures of the vessels' diameter - A1 included. Since you have not made a follow-up imaging, you have two ways to solve this: a. include the ruptured status in the multivariate analysis and discuss accordingly. Prevalence-wise, your findings will likely correlate to ruptured status. Then, discuss how this impacts the results. Or b. discuss that your methodology is insufficient to provide clear conclusions about baseline vessel assessments - therefore, you cannot exclude that the results can be biased. Accordingly, your conclusion should be that this exploratory study generates the hypothesis that longstanding A1 changes could correlate to MCA aneurysms, which should be the subject of a subsequent, more robust investigation.

Author response:Thank you for your comment. In fact, I have already discussed the relevant issues in lines 302-305 of the Limitation section. We have also made revisions to the conclusion section based on your suggestions.

3) The prevalence of A1 dysplasia in the aneurysm group was 39/269 (0.1449). The prevalence of A1 dysplasia in the control group was 29/269 (0.1078). The reported p-value (P = 0.0125) is not reproducible - neither by the two-sided Fisher nor the Chi-square tests. I found the significance levels varied from 0.2428 to 0.1945. For the suggested level of significance, you would need at least 1,263 subjects. If this is not the case, please explain and attach you statistical calculations to clarify.

Author response:Thank you very much for pointing out our significant mistake. We have rechecked the data and found that 20 was mistakenly written as 29. You can roughly estimate from the bar chart in Figure 7 that our data should be 20 instead of 29. The data in the manuscript has been corrected. We sincerely hope that you can forgive us for making such a basic error. Thank you!

3)You replied about the measurement methodology question, and cited two articles. The cited studies does not backup your claims. If you take a closer look at the Krasny (DOI:10.1136/neurintsurg-2013-010669) and Lazzaro (DOI:10.1136/jnis.2010.004358), both used planar DSA as the reference method. Interestingly, those authors address the limitations of CTA in assessing the vessels diameters. My suggestions are: a. find the appropriate references that back up your claims, but be mindful that 95% of reader are convinced that DSA is still the gold standard, as medication societies support. Or b. state that you developed this methodology and discuss the limitations.

Author response:Thank you for your comment.Allow me to explain this issue. The controversy between DSA and CTA in assessing vascular accuracy has been ongoing for a long time, with DSA being considered the gold standard. However, in clinical practice, there are many cases where there is significant external vessel diameter thickening and internal lumen narrowing. Our understanding is that DSA more accurately assesses the internal diameter of vessels, while CTA can more accurately assess the external diameter of vessels. In studying the integrity of the Wills' circle, the disadvantage of DSA is that it cannot simultaneously visualize all vessels of the circle and may not visualize vessels that actually exist but have very slow flow. Therefore, some scholars tend to use CTA to study the integrity of the Wills' circle, as mentioned in lines 215-216 of the original manuscript. We also believe in the authority of DSA, but in studying changes in vessel diameter, CTA can achieve results similar to DSA. However, we consider this is not the main focus of this article, so we only briefly touch upon it. Regarding the citation you mentioned, I borrowed the measurement method in the mid-segment 1/3 of the vessels from these two authors. Thank you!

Reviewer #9: (No Response)

Author response:Thank you!

---

## [Decision Letter · Decision Letter 2]

4 Dec 2024

PONE-D-24-36671R2Association of middle cerebral artery aneurysms and variation of the A1 segmentPLOS ONE

Dear Dr. Gao, 

Thank you for submitting your manuscript to PLOS ONE. After careful consideration, we feel that it has merit but does not fully meet PLOS ONE’s publication criteria as it currently stands. Therefore, we invite you to submit a revised version of the manuscript that addresses the points raised during the review process.

We look forward to receiving your revised manuscript.

Kind regards,

Atakan Orscelik

Academic Editor

PLOS ONE

Journal Requirements:

Reviewers' comments:

Reviewer's Responses to Questions

**Comments to the Author**

1. If the authors have adequately addressed your comments raised in a previous round of review and you feel that this manuscript is now acceptable for publication, you may indicate that here to bypass the “Comments to the Author” section, enter your conflict of interest statement in the “Confidential to Editor” section, and submit your "Accept" recommendation.

Reviewer #3: (No Response)

Reviewer #6: (No Response)

Reviewer #7: All comments have been addressed

2. Is the manuscript technically sound, and do the data support the conclusions?

Reviewer #3: Yes

Reviewer #6: Partly

Reviewer #7: Partly

3. Has the statistical analysis been performed appropriately and rigorously? 

Reviewer #3: Yes

Reviewer #6: Yes

Reviewer #7: Yes

4. Have the authors made all data underlying the findings in their manuscript fully available?

Reviewer #3: Yes

Reviewer #6: Yes

Reviewer #7: Yes

5. Is the manuscript presented in an intelligible fashion and written in standard English?

Reviewer #3: No

Reviewer #6: Yes

Reviewer #7: Yes

6. Review Comments to the Author

Reviewer #3: The authors have revised most of the comments of the reviewers, but the paper is still difficult to read for grammar and English style errors, for inconsistencies e needless repetitions.

To mention a few:

- In figures 4 to 8 the order of the columns is always different (control and aneurisms).

- M1 in figure 2 is not indicated

- The aneurism in figure 3 is indicated in the caption but not in the image.

- The short title is as long as the full title.

- “Fetal” and “embryonic” are used in an unclear way.

- Line 60: reference to Padget is not accompanied by a correct citation in the reference list at number 16.

- Line 98: The imaging scheme is still a kind of radiological recipe, unsuitable for the methods section.

- Line 104: tracker tracking?

- Line 119: calculated or rather counted?

- Line 124: A1 vessel measurement is not visible. The left and right sides should also be indicated.

- The headings of the tables are at the bottom and not above the table.

- Line 155: what is the development?

- Table 1: the rows are unclear as are their titles. Some significances are missing in the right column.

- Line 181: the diameters are not reported and there is no reference to table 2 where it is described.

- Table 4: the headings of the first column are in multiple lines and comprehension of the table is difficult.

- Line 208: researchs should be research.

- Line 235: Reference 17 is missing after Kayembe and reported in the following sentence, which refers to a different study.

- Line 287: the highest.

- Line 309: And…

Reviewer #6: After reading the revised manuscript, I have the following suggestions: In terms of the current mainstream view, the vast majority of physicians firmly believe that digital subtraction angiography (DSA) is still the gold standard for the diagnosis of cerebrovascular disease. This is especially true for issues such as cerebrovascular diameter; however, all of the data in this article are derived from CTA results, and it is recommended that this controversial issue and the limitations of CTA be discussed.

Reviewer #7: (No Response)

7. PLOS authors have the option to publish the peer review history of their article (what does this mean? ). If published, this will include your full peer review and any attached files.

**Do you want your identity to be public for this peer review?** For information about this choice, including consent withdrawal, please see our Privacy Policy .

Reviewer #3: No

Reviewer #6: No

Reviewer #7: No

---

## [Author Response · Author response to Decision Letter 3]

13 Dec 2024

Dear Editor:

We have made the modification according to your request and uploaded the additional materials as required, please check. Thank you!

Kind regards,

Hongzhi Gao

---

## [Decision Letter · Decision Letter 3]

15 Jan 2025

PONE-D-24-36671R3Association of middle cerebral artery aneurysms and variation of the A1 segmentPLOS ONE

Dear Dr. Gao,

Thank you for submitting your manuscript to PLOS ONE. After careful consideration, we feel that it has merit but does not fully meet PLOS ONE’s publication criteria as it currently stands. Therefore, we invite you to submit a revised version of the manuscript that addresses the points raised during the review process.

We look forward to receiving your revised manuscript.

Kind regards,

Atakan Orscelik

Academic Editor

PLOS ONE

Journal Requirements:

Additional Editor Comments (if provided):

Reviewers' comments:

Reviewer's Responses to Questions

**Comments to the Author**

1. If the authors have adequately addressed your comments raised in a previous round of review and you feel that this manuscript is now acceptable for publication, you may indicate that here to bypass the “Comments to the Author” section, enter your conflict of interest statement in the “Confidential to Editor” section, and submit your "Accept" recommendation.

Reviewer #3: (No Response)

Reviewer #6: All comments have been addressed

2. Is the manuscript technically sound, and do the data support the conclusions?

Reviewer #3: Yes

Reviewer #6: Yes

3. Has the statistical analysis been performed appropriately and rigorously? 

Reviewer #3: Yes

Reviewer #6: Yes

4. Have the authors made all data underlying the findings in their manuscript fully available?

Reviewer #3: Yes

Reviewer #6: Yes

5. Is the manuscript presented in an intelligible fashion and written in standard English?

Reviewer #3: No

Reviewer #6: Yes

6. Review Comments to the Author

Reviewer #3: The authors have accepted the comments of the previous revision and have corrected the paper accordingly. Yet, they failed to carefully revise the whole paper to avoid inconsistencies and style imperfections.

To mention a few:

- Line 98: start the sentence like this: “CTA was performed with a 64-row helical CT scanner (PHILIPS or TOSHIBA AQUILION 64) following the standard protocol at ….”.

- Line 116: “Figure 3: Example of image reconstruction. The single arrow indicates dysplasia of the A1 segment of the left anterior cerebral artery, while the double arrow indicates a MCA aneurysm. Abbreviations as in figure 2”.

- Line 129: Wills ring!

- Line 133: remove “Scatter plots displayed the”.

- Line 135: remove “Bar graph of”.

- Line 137: same as above.

- Line 152: same as above. Furthermore, caption of figure 7 indicates dysplasia, while in the drawing it is indicated as “hypoplasia”.

- Caption of figure 8 is missing.

- Table 1: the outline is misleading. The statistical analysis for PcoA and PCA was presumably done with chisquared. Therefore, the relevant lines should be closer to provide visual indication of the single p. From my calculations, p=0.6031 for PcoA is wrong.

- The term hypoplasia should have been removed, but it is still used many times from line 165 and the following.

- Line 195: use variations instead of deficiencies.

- Line 250: reference number (47) is too far away from Fahy et al. Put it at the end of the same sentence.

- Line 255: same as above for Lazzaro et al.

- Figure 4: age is not centered on the Y ax on the left.

Reviewer #6: After reading the author's revised draft, the author responded to the queries raised in a way that did not convince me, but did not detract from its publication as a paper.

7. PLOS authors have the option to publish the peer review history of their article (what does this mean? ). If published, this will include your full peer review and any attached files.

**Do you want your identity to be public for this peer review?** For information about this choice, including consent withdrawal, please see our Privacy Policy .

Reviewer #3: No

Reviewer #6: No

---

## [Author Response · Author response to Decision Letter 4]

19 Jan 2025

Dear Editors,

On behalf of my co-authors, we thank you for giving us the opportunity to revise and improve the quality of our article. We have carefully reviewed the comments from both the reviewers and yourselves, and have made revisions which are marked in red within the paper. We have made every effort to address the comments provided. Please find attached the revised version, which we would like to submit for your kind consideration.

In accordance with your request, I have ensured that the citations are accurate and complete, and that no references have been made to retracted papers. Thank you.

We have prepared a detailed response letter addressing each of the reviewer comments, which can be found at the end of this letter. We found the comments to be extremely helpful and would like to express our gratitude to both you and the reviewers for your valuable feedback.

We hope that this revised version meets the standards for publication in your journal. Thank you for considering our submission, and we look forward to hearing from you soon.

Sincerely,

Hongzhi Gao

Reviewer #3: The authors have accepted the comments of the previous revision and have corrected the paper accordingly. Yet, they failed to carefully revise the whole paper to avoid inconsistencies and style imperfections.

To mention a few:

- Line 98: start the sentence like this: “CTA was performed with a 64-row helical CT scanner (PHILIPS or TOSHIBA AQUILION 64) following the standard protocol at ….”.

- Line 116: “Figure 3: Example of image reconstruction. The single arrow indicates dysplasia of the A1 segment of the left anterior cerebral artery, while the double arrow indicates a MCA aneurysm. Abbreviations as in figure 2”.

Author response:Thank you for your comments, which have been revised according to your comments.

- Line 129: Wills ring!

Author response:Thank you for your comments, which have been revised according to your comments.

- Line 133: remove “Scatter plots displayed the”.

- Line 135: remove “Bar graph of”.

- Line 137: same as above.

- Line 152: same as above. Furthermore, caption of figure 7 indicates dysplasia, while in the drawing it is indicated as “hypoplasia”.

Author response:Thank you for your comments, which have been revised according to your comments.

- Caption of figure 8 is missing.

Author response:Thank you for your valuable advice. I apologize for the error. Figure 8 title has been added.

- Table 1: the outline is misleading. The statistical analysis for PcoA and PCA was presumably done with chisquared. Therefore, the relevant lines should be closer to provide visual indication of the single p. From my calculations, p=0.6031 for PcoA is wrong.

Author response:Thank you for your comments, which have been revised according to your comments.

- The term hypoplasia should have been removed, but it is still used many times from line 165 and the following.

- Line 195: use variations instead of deficiencies.

- Line 250: reference number (47) is too far away from Fahy et al. Put it at the end of the same sentence.

- Line 255: same as above for Lazzaro et al.

Author response:Thank you for your comments, which have been revised according to your comments.

- Figure 4: age is not centered on the Y ax on the left.

Author response:Thank you for your valuable comments and corrections have been made to the chart.

Reviewer #6: After reading the author's revised draft, the author responded to the queries raised in a way that did not convince me, but did not detract from its publication as a paper.

Author response:Thank you!

---

## [Decision Letter · Decision Letter 4]

4 Feb 2025

Association of middle cerebral artery aneurysms and variation of the A1 segment

PONE-D-24-36671R4

Dear Dr. Gao,

We’re pleased to inform you that your manuscript has been judged scientifically suitable for publication and will be formally accepted for publication once it meets all outstanding technical requirements.

Kind regards,

Atakan Orscelik

Academic Editor

PLOS ONE

Additional Editor Comments (optional):

Reviewers' comments:

Reviewer's Responses to Questions

**Comments to the Author**

1. If the authors have adequately addressed your comments raised in a previous round of review and you feel that this manuscript is now acceptable for publication, you may indicate that here to bypass the “Comments to the Author” section, enter your conflict of interest statement in the “Confidential to Editor” section, and submit your "Accept" recommendation.

Reviewer #3: All comments have been addressed

2. Is the manuscript technically sound, and do the data support the conclusions?

Reviewer #3: Yes

3. Has the statistical analysis been performed appropriately and rigorously? 

Reviewer #3: Yes

4. Have the authors made all data underlying the findings in their manuscript fully available?

Reviewer #3: Yes

5. Is the manuscript presented in an intelligible fashion and written in standard English?

Reviewer #3: No

6. Review Comments to the Author

Reviewer #3: The authors have amended the paper that is now more intelligible. However, there are still a few changes easy to apply:

- line 196: remove "while"

- line 203: remove "however"

- line 241: change "dysplasia" to "dysplastic" or "have dysplasia"

- line 262: "Willis"

- line 275: change "dysplasia" to "dysplastic" or "have dysplasia"

7. PLOS authors have the option to publish the peer review history of their article (what does this mean? ). If published, this will include your full peer review and any attached files.

**Do you want your identity to be public for this peer review?** For information about this choice, including consent withdrawal, please see our Privacy Policy .

Reviewer #3: No

---

## [Editor Report · Acceptance letter]

PONE-D-24-36671R4

PLOS ONE

Dear Dr. Gao,

I'm pleased to inform you that your manuscript has been deemed suitable for publication in PLOS ONE. Congratulations! Your manuscript is now being handed over to our production team.

Kind regards,

on behalf of

Dr. Atakan Orscelik

Academic Editor

PLOS ONE